# Mutual Doings: Exploring Affectivity in Participatory Methodologies

**Karin Gunnarsson**

Department of Education, Stockholm University, 114 19 Stockholm, Sweden; karin.gunnarsson@edu.su.se

**Abstract:** The aim of this paper is to explore the affective implications of working with participatory methodologies within the context of sexuality education. For this exploration, a feminist posthumanist approach is put to work, building on a relational ontology and the notions of affectivity, assemblage and environmentality. Drawing from a practice-based research project concerning sexuality education conducted together with teachers in Swedish secondary schools, the analysis puts forward how the research assemblage navigates and manages affective conditions in ways that produce, allow and exclude certain feelings. With (dis)trust, uncertainty, frustration, laughter and shame, the assemblage made bodies act and become in specific ways. Thus, the analysis shows how participatory and practice-based research become moulded by power relations and intense flows of desire working together. This raises questions about how participatory methodologies within an ontological view of interdependence afford to manage affective intensities to move in certain directions of socially just sexuality education.

**Keywords:** participatory methodologies; sexuality education; affectivity; feminist posthumanisms; assemblage; environmentality

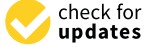



## 1. Introduction

Within the context of sexuality education, this paper tackles the encounter of participatory methodologies and affectivity. As a vignette, we will visit a workshop with 12 secondary school teachers and three researchers as part of a research project focusing on sexuality education. At the time of this workshop, the teachers and researchers had collaborated for one term and knew each other quite well. A teacher in foreign languages, David, tells the group about an experience of showing his German class a school-allocated episode of a specific television series:

> David: In the ninth grade, I showed 'Skam Deutschland', but I felt my hair stand on end, so I had to turn it off.
>
> Anna: Why?
>
> David: Well, it was so very intimate, like...
>
> Laughter
>
> David: But, oh my god, I had to put it away and say that this is school television so they wouldn't think that it was chosen by me.
>
> Laughter
>
> David: I don't know what it was, but it was twice as bad, or as intimate, as the Spanish version, and I realised that way too late. It was a Thursday afternoon class, and I thought, now we can watch this; but it just got worse and worse, and I thought, what should I do? So, I said, now I really think we should take a break. But they liked to watch it. [Turning to the researcher] Hope you didn't write everything down.

Even though, since the 1950s, in Swedish schools, sexuality education has been mandatory, it is still regarded as a risky business. As a complex and sensitive knowledge area, it has been marginalised with low status in relation to the core curriculum. Its fussy character, lack of teacher training and competing educational priorities make sexuality education a difficult task for schools to handle. However, the field of sexuality education has recently gained renewed interest in policy and practice contexts in Sweden. With a revised curriculum, including reinforced formulations concerning what is labelled the knowledge area of sexuality, consent and relationships (Swedish National Agency for Education 2022), many teachers, researchers, interest groups and policymakers find it urgent to explore how to carry out sexuality education. Thus, this creates an opportunity to rethink and reimagine both what sexuality education might become in the school setting and how this can be done by working with participatory methodologies.

With this educational–political setting as a point of departure, four colleagues and I conducted a practice-based research project concerning sexuality education in secondary schools. The project had the ambition of collaborating with teachers to reimagine what is both 'doable (pragmatic) and possible (speculative)' (Renold et al. 2021, p. 543). This meant experimenting with problems rather than seeking answers or solutions. As can be seen in the introductory example, the area of sexuality education is saturated with affectivity. When the teacher, David, shares the story about showing his class a school television programme with what is described as too intimate content appropriate for teaching, the spatial atmosphere becomes loaded with joy, excitement and embarrassment. In other words, within this event, specific affective conditions emerge. Laughter becomes a forceful actor as it accentuates a sense of attachment and recognition when stressing how the ambiguous character of sexuality can be difficult to handle. Additionally, the ongoing research becomes another co-producing aspect, as David ends the story by commenting on the situation being documented altogether, creating an affective-spatial assemblage with vital implications for the knowledge-production of both the research and teaching at hand.

Drawing from the project, this paper explores the affective implications of working with participatory methodologies within the context of sexuality education. This raises questions on how to consider the affective conditions intrinsically involved when working with participatory methodologies. These questions will be explored with a feminist posthumanist theoretical framing that 'turns the attention to sensuous, affective, material and spatial qualities' (Juelskjær 2017, pp. 65–66), co-creating both educational and research practices. Subsequently, this paper aims to theoretically and empirically explore how participatory methodologies involve affectivity. Moreover, the paper aims to illustrate how sexuality education offers specific affective conditions for our collective work. The research questions include: how does affectivity induce the conditions of participatory methodologies, and how to address the (im)possibilities of the indeterminate character of affectivity in this setting?

To answer the aim and questions, the article unfolds as follows. The upcoming section outlines research concerning participatory methodologies as well as research on sexuality education and affectivity, specifically with a feminist posthumanist approach. Next, the theoretical take on feminist posthumanism and the practice-based approach employed are described. After that, the analytical exploration of affective conditions is based on empirical research events when working with participatory methodologies. To conclude, the potentialities of acknowledging affectivity in participatory methodologies are discussed.

## 2. Participatory Methodologies, Affectivity and Sexuality Education

To situate the paper, this section addresses educational research grounded in feminist posthumanist theories working with participatory and practice-based methodologies. It also pays specific attention to research on sexuality education as well as affectivity. What this brief review of educational feminist posthumanist research is set to address is the inspiration it carries towards exploring the methodological and empirical implications of affectivity.

Grounded in various epistemologies and ontologies, educational participatory methodologies include a range of different approaches, for example, practice-based research, participatory action research and critical practitioner research (see Candy et al. 2022). What these various approaches have in common is an interest in how research can be done in collaboration with participants and practices and, as such, acknowledge how doing research means intervening and collaborating. With a democratic ethos, research participants are regarded here as more than mere informants that emphasise how knowledge production emerges with and not only on or about the research participants (Gunnarsson 2018).

In the last decade, feminist posthumanist theories have significantly influenced educational research and have repeatedly stressed the significance of participatory and practice-based methodologies (Duggan 2021; Mörtsell and Gunnarsson 2023; Niccolini and Ringrose 2019; Renold and Timperley 2023). Within the feminist posthumanist framework, epistemology, ontology and methodology are seen as interdependent, which means that knowledge practices are performative and cannot be separated from world-making practices. Therefore, the methodological literature on educational posthumanism calls for acknowledging the messy and performative practices of knowledge production enacted together with the world (Mörtsell and Gunnarsson 2023). This is described as playing with 'methods that are not pre-planned or predetermined before the encounter but are instead *emergent with* the research' (Weaver and Snaza 2017, p. 1063, italics in original).

In sexuality education research, scholars are also pursuing posthumanist-inspired participatory methodologies (see, e.g., Gunnarsson and Ceder 2023; Renold et al. 2021; Renold and Timperley 2023). A research-activist inquiry is endorsed here, stressing how creative engagement with the world is crucial to making sexuality education, as well as educational research, matter (Renold et al. 2021; Ringrose et al. 2019). With this approach, the 'response-abilities' of doing research imply 'working as collaborative assemblages in order to generate social changes' (Ringrose et al. 2019, p. 262). Furthermore, many scholars within a posthumanist framework acknowledge how affectivity is an integral dimension of both research and educational practices (e.g., Dernikos et al. 2020; Gunnarsson 2022; Zembylas 2022). These scholars underline the relational aspect of affectivity as well as its entanglement with materiality and spatiality to accentuate 'the affective-material life of the spaces we teach and research in' (Niccolini et al. 2018, p. 324).

Collectively, the array of educational participatory and feminist posthumanist studies accounted for here push the conceptions of collaboration, intervention and practices. As such, they help address the co-producing aspects of research. Then, this paper seeks to contribute to the literature through an interplay of empirical, theoretical and methodological exploration focusing on affectivity. In doing this, the contribution implies a reconfiguration of how affectivity co-creates our research practices and the impact it has in its unpredictable feature of 'happening all around us' (Dernikos et al. 2020, p. 19).

## 3. Feminist Posthumanisms and Relational Ontology

As mentioned, the theoretical take for exploring affectivity in participatory methodologies is grounded in feminist posthumanisms and a relational ontology (Braidotti 2021). With this, it becomes possible to explore how participatory methodologies involve affective conditions entangled with bodies, things, ideas, words, etc. Primarily, the notion of assemblage addresses the ontological account of the contingent character of affectivity, materiality and discourses. This means that affectivity is constituted and acquires its characteristics within the relationality of the assemblage (Mörtsell and Gunnarsson 2023).

Herein, I follow the post-Deleuzian take on affectivity as collective bodily and spatial generative forces that produce connections and enactments outside the control of humans alone (Braidotti 2017, 2021). Affectivity does not only concern someone or something but also co-creates the conditions of places, bodies and doings. Then, this implies encountering affectivity within two interwoven senses, both as indeterminate conditions creating the capacity to affect and be affected and as feelings circulating in particular spaces, such as classrooms. Accordingly, affectivity is regarded as intensities that both move and structure

what takes place (cf. Dernikos et al. 2020; Gunnarsson 2022). Accordingly, within this Deleuzian view, affectivity is conceptualized as forces of desire continuously moving and making connections. By escaping consistency, affectivity, desire and power are intrinsically connected in their capability of creating unpredictable directions (Braidotti 2021).

Additionally, in this discussion, the theorising on affectivity is connected to the work of Juelskjær and Staunæs (2016) and their notion of 'environmentality'. Environmentality entails mingling the Foucauldian concept of governmentality with affectivity. This implies that 'environmentality is a specific sort of governing that manages the intensities instead of identities and does so through modulation of the environment and by facilitating possible fusions, openings and connections' (Juelskjær and Staunæs 2016, p. 188). Then, environmentality actualises collective adjustments that manage and stage intensities and addresses a shift from governing identities 'towards orchestrating the affective intensities' part of producing educational spaces and subjectivities (Juelskjær and Staunæs 2016, p. 184). In managing intensities, environmentality is not considered good or bad but instead becomes a way to address how affectivity is inevitably and collectively governed within a 'complex topology of power' (Juelskjær 2017, p. 81).

Then, working with the notions of affectivity, assemblage and environmentality makes it possible to consider intensities, frictions and movements in terms of how they operate as well as how they are managed in a manner contingent upon relational doings. Reorienting participatory methodologies with the help of this framework disrupts subject-centred conceptions and acknowledges the affective conditions enmeshed when doing research. As such, this paper addresses how participatory methodologies involve concerns about both being sensitive to and working with the cultivation of affective conditions.

## 4. Methodology: Collaborative Doings and Engagements

Conducting participatory research with a feminist posthumanist approach makes it possible to acknowledge how affectivity emerges in joint practices. By emphasising doings and engagement, exploring affectivity with this approach indicates interfering and involving in practices, such as teaching, and experiencing its forces, difficulties and joy. Then, this means to address how the research and researcher are always 'entangled and emergent with other beings and things' (Weaver and Snaza 2017, p. 1063). To do this, the research project strived to create collaborative engagement around the problem of arranging the teaching of sexuality education. More specifically, the project was concerned with how to improve sexuality education as a transdisciplinary knowledge content part of ordinary classroom teaching, targeting primarily teachers at different school subjects as carters of the process to create sustainable change (cf. Ollis and Harrison 2016). Accordingly, the collaboration with teachers entailed intervening in embedded and embodied curiosities, not only to investigate how sexuality education was done but also to work together to push collectives into new directions.

In the participatory and practice-based research project, we were five researchers and former teachers involved in collaborations with 5–13 teachers at different school subjects at four secondary schools. The collaboration included 5–8 workshops with teachers that took place mainly in the respective school but also at the university and via Zoom due to the pandemic. The empirical material analysed in this paper was then constructed through the workshops using audio recordings and field notes. Research ethics were considered throughout the research process, in the empirical as well as analytical work, and all names are pseudonyms.[1]

Analysing the empirical material within feminist posthumanism allows for exploring and reconceptualising how participatory methodologies are part of co-constructing affective conditions. As affectivity is a messy component challenging to bear or linger, the analysis was done by slowing down and becoming attentive to events enmeshed with modes of intensity and friction. This meant unfolding empirical events sensitive to how the research practice collectively arranged and managed affective assemblages.

*Introducing Collaborative Doings*

Before analysing the empirical events, I will linger further on the collaborative doings in terms of the workshops. The workshops emerged from collaborations between researchers and teachers and were both exploratory and pedagogical in character. This meant that the various things we did together were simultaneously considered research and teaching methods (Lenz Taguchi et al. 2020). As such, they responded out of a joint engagement in exploring how to tackle sexuality education. In this sense, they could be described as a subtle form of posthuman pedagogy or making together to set sexuality education in motion and make it matter (cf. Niccolini and Ringrose 2019).

Each school's initial workshop started with a discussion about what sexuality education could include, simultaneously manifesting and contesting its messy features. An inventory exercise followed this to explore what was being done at the school and to map out the ambiguous character of sexuality education everywhere (Renold et al. 2021). After that, the workshop was followed by discussions on what topics the teachers found vital to continue working on to guide the content of the upcoming meetings. We, as researchers, were in charge of the planning, but the content was decided on together with the teachers. This meant recurrently checking with the teachers' concerning topics that troubled them, for example, by ending the workshops with discussions and writings on what to work on at the upcoming meeting and how to approach those ideas. In our collective work, affectively loaded questions, such as pornography, gender diversity, the body and norm-critical pedagogy, were discussed (cf. Gunnarsson and Ceder 2023). This was done together with teaching materials, including pictures, films and exercises. Critically and creatively, we engaged with the teaching materials, discussing if and how they could be involved in teaching and what could take place in relation to them. However, the planning also involved a degree of tension for us as researchers. These tensions can be articulated as being concrete but not too supervisory, academic but not too theoretical, close to the teachers' everyday doings, but at the same time, pushing for possibilities.

Hence, the participatory and practice-based research approach meant that we, in the research group, in relation to sexuality education, the schools and teachers, by some means, staged the affective-spatial conditions to create engagement. Although regarding the indeterminacy of affectivity, the foundation of our collaboration was built on the desire to create affective connections to push the boundaries of what sexuality education could become (cf. Gunnarsson 2021, 2022; Zembylas 2022)—to push boundaries that make certain bodies, peoples and knowledge feel out of place in the teaching of sexuality education. Now, we will turn to the empirical events where this was carried out.

## 5. Events of Affective Intensities

This section explores events where affectivity was specifically manifested to address its co-producing effects in participatory methodologies. Needless to say, the collaborations turned out differently at the three schools, but the overall experience was that the collaborations were productive even though there were tensions and quarrels. However, the collaborations all started with a joint ambition to revise how sexuality education was done.

### 5.1. Doing Something and Doing It Right

At one of the schools, the initiative to join the research project came primarily from one science teacher, Fredrik. Fredrik expressed that he was alone in the work of sexuality education, and by being part of the research project, he now felt allocated to do something extensive at the school.

> Fredrik: The goal for me is that we have to do something concrete. We have to reach high; we must have a theme week, that should be the goal. Now we have an ongoing project, which means we get another mandate to enter the school's activity and manage a whole week.
>
> Louisa [researcher]: What do the rest of you say?

Malin: I think it sounds good. I also want to reach high, especially since the school has joined this project; it has to lead to something, at least three days.

Here, the teachers express their aspirations as well as their expectations of how the collaboration should result in something distinct and substantial. In relation to being part of the research project, they see an opportunity to make an impact on the school. With enthusiasm and determination, the teachers become eager to change the sexuality education programme, and it becomes important to set up directions for achieving a theme-based week. In this sense, the collaboration involved affective conditions of trust together with longings for a concrete, manageable and effective process. The research project became part of creating capacities for the teachers to highlight sexuality education in relation to colleagues and school administration. As such, it became an actor involved in making sexuality education important and putting it on the school agenda.

Thus, this can be understood as navigating the affective implications of the logic of schooling and academia ingrained in the very fabric of historical–political power formations. Entwined with the logic of improvement and effectiveness, the collaboration involved affective traces that had to make an impact and lead to something. As Renold et al. (2021, p. 540) state, we were all part of 'hyper-rationalist "what works"' rationales co-creating affective conditions for the collaboration. For us researchers, these rationales were also co-producing uncertainty about how to create collaborations that could be considered relevant and evocative—creating worries that it had to be worth the time the teachers spent with us working with sexuality education.

At one of the other schools, the expectations of the workshop had a different character. The teachers had a tradition of conducting a theme-based week on sexuality education and now wished to focus on reconsidering its content by amplifying collaborations and progression. At one of the workshops, the difficulties of teaching sexuality education and handling these topics with their students became stressed, as they were working in a setting where racism, sexism and homophobia were recurrently present. This urged one of the researchers to ask about what could be considered doable.

Tina [researcher]: I have to ask this question, is this kind of discussion impossible to have with your groups [of students]?

Stefan: No, it's not impossible.

Filippa: Then we wouldn't be sitting here.

Stefan: In the vast majority of cases, it works fine. We are talking about the situations when it doesn't work, and these involve a few students.

Filippa: We talk about our worries and what we want to learn to be able to handle those situations as well.

Herein, the collaboration created a space moulded with feelings of despair when stressing worries and troubles. Engaging with the troubles the teachers face in a violent classroom, Tina raises a provocative question about the impossibility of specific discussions with the students' concerning issues related to sexuality education. The question navigates the affective condition and brings forward statements such as 'then we would not be sitting here' and 'we want to learn'. As such, it pushes the discussion in new directions, transforms the affective conditions from getting stuck in despondency and, instead, cultivates the assemblage into something productive by unfolding the expectations and desires for our collaboration.

Thinking about this event with the notion of environmentality suggests how collaboration becomes an affective-spatial learning assemblage (Juelskjær and Staunæs 2016, p. 193). In terms of wanting to know how to handle difficult situations, the adjustment that the question provokes transforms the affective conditions into an intensification of learning. Infused with the logic of what works, the assemblage encourages solutions rather than 'staying with the troubles'. Asking about the (im)possibilities can then be seen as a way of managing the intensity where the question effectively moved bodies into how to

navigate the troubled life world of the classroom. Moreover, this event highlights how the affective-spatial research assemblage involves affirmation and critique within the discomfiture and affectively charged doings. As such, the governing of affects constructs multifaceted feelings of trust, worries and hopes.

These two events unfold how participatory methodologies emerge within the affective condition of uncertainty regarding what to do but at the same time encompass a trusting sense of transformation. This means that engaging in participatory methodologies involves the insecurity of not knowing where it is leading but 'creating space for what might take place in the encounter' (Gunnarsson 2021, p. 72). Within the limited possibilities of knowing what will happen or the right solutions, collaboration instead involves a process of experimentation with a responsibility to be responsive to the moment (cf. Gunnarsson 2018; Renold et al. 2021). This means navigating the tensions of orchestrating affective intensities and trusting the process and encounters to afford careful and mutual doings.

*5.2. Multiple Co-Producers*

As described earlier, at the first workshop, we discussed what sexuality education implies to grasp its fussy and moving character. Asking if it could be everything in order to collaboratively unfold and push its borders and challenge, for example, dominating risk-infused medical discourses (Gunnarsson 2023; Cameron-Lewis 2019). We started the exercise in smaller groups of two or three, taking notes on Post-its. Then we gathered the whole group to create an overview of the different things happening. At one of the schools, this exercise produced an intense atmosphere.

> Malin: I usually show this movie *Sex on the Map*. It's a bit old, but I think it's good. It's an animated movie that I show during the mentoring time.
>
> *Biology teacher Fredrik mumbles and looks at the floor, which evokes some laughter.*
>
> Malin: What is it? Are you laughing behind my back?
>
> Fredrik: Yes, I just, like, this, huh? You, just, 'I usually show it during the mentoring time'. Can't we then show it in connection with sexuality education?
>
> Malin: Yes, but if you do that—I usually ask the class, have you seen this film? And they answer 'no'. Are you done with the sexuality education? Do you want to see it? They say, 'yes', and then they get to see it. But you show it?
>
> Fredrik. Yes, of course.
>
> Nadia: I would think like this, it is very interesting for the students in sexuality education; maybe they want to see it one more time?
>
> Malin: Yes.
>
> *Laughter*
>
> Johan [researcher]: This is exactly why it is so good when you do something like this because then you can coordinate it. The next step would be to address something you do while teaching during the mentoring time.
>
> Malin: It can't be so that you own these things in biology to talk about, Fredrik?
>
> *Fredrik points to the recorder at the table and asks:* 'Can I turn this off?'
>
> *Laughter*

When Malin tells the group that she shows the movie *Sex on the Map,* Fredrik reacts by questioning why this is not done in relation to sexuality education. Here, the fussy character of sexuality education emerges, creating tensions about where and how it should take place. Moreover, the event co-created the dominant discourse that is mainly situated within the school subject of biology. Something that is questioned by Malin's frank statement about who or rather which school subjects 'own these things'.

This conversation evoked a tense atmosphere, intermingling frustration, intimidation and laughter—frustration of not knowing what is being done at the school and not having the opportunity to cooperate and coordinate around the issues. The affective intensity

moves in different and contradictory directions, where both a teacher and a researcher make comments about staging a collective understanding and resolving the emerging conflict rather than remaining in frustration, emphasising what can be learned from the situation and how it can be regarded as productive for the everyday doings of sexuality education.

Furthermore, the research practice becomes emergent when Fredrik indicates that the conversation is being recorded. This highlights the recorder as an influential affective actor in regulating what can and cannot be said. As such, the recorder becomes part of producing the situation and emphasises the affective intensity within the research assemblage. The teachers' and researchers' bodies, the space we collaborate with and engage in, and the very materialities that are co-creating them involve a multitude of affective histories, possibilities and trouble within the assemblage of the classroom, students, film, time, research, school subjects, etc.

How teaching materials, such as films, became affectively loaded also became apparent in the next event. Here, one of the teachers questioned a film we, the researchers, had suggested they watch before the workshop.

> Filippa: It was five years ago that we talked about this norm-critical approach. The film just showed things that don't work, and we already know that things can turn out that way. I don't feel like I'm getting anything new out of it, but maybe I'm missing something.
>
> Tina [researcher]: More comments?
>
> Helena: Well, I still think it's quite refreshing to repeat this with your own 'luggage' and what you bring with you. You have definitely learned it somewhere, but then the every day comes, so you don't think about it. [. . .] I think it's quite good because there are often things that go wrong, so they [students] make mistakes, and I make mistakes, and then we work through it.
>
> Filippa: Yeah, but you didn't have to watch this video to do that. I think I want something new and something more.
>
> Helena: Absolutely, I agree with that, but I still thought it was nice to have repetition.
>
> Filippa: That sounds good; repetition is always good.

In this event, the teacher Filippa asserted that the film about norm-critical pedagogy did not bring anything new but only repeated what was already known. Moreover, it foremost underlined the things that can go wrong. As a response, Helen says that it was valuable as a reminder of the difficulties of everyday teaching and how mistakes can create discussions and resolution. Filippa agrees with the positive effects of repetition but asks for something new. The researcher then handles the situation by asking for more comments instead of making arguments or defending the movie. Within a cautious researcher position, the affective conditions are navigated by restraining the precarious situation. By attuning to the discussion, the research assemblage carefully considers the feelings of not getting answers or solutions. This implies managing the affective conditions and avoiding clashes between researchers and teachers for productive encounters and joint doings.

This event raises questions about the affective conditions of what could be deemed new and how our learning affects everyday teaching practices. This means that affective conditions intermingle with pasts and futures, personal and collective histories and doings (Juelskjær 2017). As such, affective conditions highlight the difficulties of creating 'meaningful social change' (Ringrose et al. 2019, p. 265) for all participating parties. But still, participatory methodologies, in their means to create collective engagement, afford to interrupt flow and energies and therefore involve both movements and stabilisations.

At another school, discussions emerged surrounding how and where to share what was being done regarding sexuality education. The discussion concerned sharing teaching materials within a range of school subjects to create collaborations around joint topics.

> Maria: Our meetings are full, but it would be good to take some time to talk about this [sexuality education] for at least five minutes.
>
> Tanya: It can be nice to have it [sexuality education] physically on the wall as a reminder. You know, we have a hundred channels and I always wonder where I should enter. Then it doesn't get done.
>
> Stefan: We also need something to liven up our room, something that would give it some feeling.
>
> Johan [researcher]: If you find an article, print it out and hang it on this [silent pause] sex wall.
>
> Tanya: Exactly, a sex wall!
>
> *Laughter*
>
> Johan: Or relationship wall or love wall. . .

At this event, with enthusiasm, the teachers urge a place to highlight what they do in sexuality education. With limited time in their weekly meetings, assigning a site in their workplace is suggested and described as making sexuality education easy to access. This implies making and giving sexuality education a space to be an influence in the ordinary life of teaching. The researcher Johan offers the suggestion of posting articles on a so-called 'sex wall'. This 'sex wall' label becomes affectively charged and provokes excitement and laughter within our collaboration.

These events disclose how materialities such as films and walls collectively and affectively animate bodies and minds in various ways. They involve the creation of specific power formations where frustration, insecurity and laughter become vigorous for teachers and researchers. In other words, participatory and practice-based research become part of the mattering of affective-spatial intensities, not in a one-way direction but rather within a reciprocal relationality of the teaching-research assemblage where 'each body is ethically entangled in the process of that worldmaking' (Ringrose et al. 2019, p. 337).

*5.3. Power and Desire Working Together*

In summary, analysing the research events wherein affectivity was specifically manifested gives a compelling reminder of the inherent unpredictability and infinite possibilities embedded within our mutual doings (Dernikos et al. 2020). By tackling sexuality education, the research collaborations highlighted what can be regarded as a sensitive and sensuous dimension of this matter. Hence, staying with affects embraced the messiness of what takes place and highlighted how affectivity becomes a vibrant actor in the web of relations. With (dis)trust, uncertainty, frustration, joy, curiosity and shame, the research assemblage made bodies act and become in different ways. What has then unfolded is the complex feature of affectivity, capable of intertwining desire and power, proximity and distance and self and others. Notably, when the research collaboration became part of orienting toward neoliberal-infused feelings of a productive and positive future, paying attention to how affectivity is collectively managed opened up the possibility to critically engage with the power formations at hand. Thus, the research apparatus became moulded by power relations and intense flows of desire working together. Within this contradictory and unpredictable setting, the relational obligation of interfering with practices became apparent. Accordingly, participatory methodologies imply a critical and creative endeavour as they co-produce affectivity as indeterminate and emergent forces that can be done differently.

## 6. Mutual Doings within Affective Conditions

Grounded in feminist posthumanism, this paper started by asserting that participatory methodologies are imbued with affectivity, requiring specific attention. The aim was then to explore the affective implications of working with participatory methodologies within the context of sexuality education, and the guiding questions were as follows: how does affectivity induce the conditions of participatory methodologies, and how can we address the (im)possibilities of the indeterminate character of affectivity in this setting? In

response to these questions, the analysis reveals how affective conditions were cultivated and brought into play, with regulatory as well as transformative effects. Within collective adjustments, the research assemblage produced, allowed and excluded certain feelings. Thus, participatory methodology became part of governing affectivity when trying to, or even longing for, foster smooth and productive collaborations. This can also be described as the methodology being part of navigating affectivity to constitute new assemblages and new doings, particularly within sexuality education.

The analysis might not yield straightforward and definitive answers. Nonetheless, it contributes with elaborations and examples of how affectivity impels the knowledge production taking place when working with participatory methodologies. By challenging assumptions about linearity and rationality, the analysis underscores the tensions and transformations enacted within the dynamic interplay of affect and materiality. What comes to the fore are the very embodied structures of knowledge production, where knowledge becomes an effect of affective processes as an "ongoing experiment with intensities" (Braidotti 2017, p. 26). While it is not the intention to claim methodological innovation, there were moments within the participatory research approach where the collaboration afforded a space to try new things—a capacity to push the boundaries of the (im)possibilities in research and teaching. Therefore, I want to stress that every minute relationship involves transformation and that feminist posthumanist participatory methodologies striving for embodied engagements and collaborations can make significant contributions to, in this case, sexuality education. Thus, I argue that highlighting affectivity expands participatory and practice-based research by disclosing how intensities co-construct collaboration and intervention. This urges us to closely interrogate how affectivity operates in, through and with the research assemblage in order to embrace "one's interconnections to others in their multiplicity" (Braidotti 2017, p. 26). Then, it will be possible to reimagine key dimensions of participatory methodologies such as power, body, agency and change. Expanding our understanding of participatory methodologies to encompass affectivity affords us to recognise how knowledge production is contingent upon the subversive power dynamics taking place and has the potential to create open and undetermined research practices.

In light of these considerations, I see that there are vital concerns left to consider regarding how affective-spatial intensities constitute movements and affect power formations within the research assemblage. This raises questions about how participatory methodologies founded on an ontological take on interdependence afford to manage affective intensities to move in certain directions of socially just sexuality education. How could managing affectivity be acknowledged in relation to trusting collaborations? What does affectivity do and be made to do in the research assemblage, and what new doings might research afford to invite and invent? Addressing these questions entails an avenue for critical and creative explorations within the realm of participatory methodologies, as we seek to embrace affectivity within our mutual doings.

**Funding:** This work was supported by the Swedish Research Council under Grant 2019-03962.

**Institutional Review Board Statement:** The study was conducted in accordance with the Declaration of Helsinki, and approved by the Swedish Ethical Review Authority (2020-00823, April 2020).

**Informed Consent Statement:** Informed consent was obtained from all subjects involved in the study.

**Data Availability Statement:** Data is unavailable due to ethical restrictions.

**Acknowledgments:** My warmest thanks to the teachers participating in the project.

**Conflicts of Interest:** The author declares no conflict of interest.

## Note

<sup>1</sup> The study underwent an ethical review and was approved by the Swedish ethical review authority (2020-00823, April 2020).

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
