# Peer review of "Mutual Doings: Exploring Affectivity in Participatory Methodologies"

_humanities, doi:10.3390/h12060131_

Round 1

Reviewer 1 Report

Comments and Suggestions for Authors

The author has provided a commendable contribution to the advancement of participatory methodology within the research community. However, there are specific points that need further attention and clarification:

1. The initial phrase in the abstract, "With a feminist posthumanist approach," appears to be incomplete. It is suggested to either omit this or furnish a comprehensive problem statement as a precursor to the objective/aim.

2. Regarding construct number 6 "Mutual doings within affective conditions", it should not be presented as the conclusion. Instead, it represents the primary finding or argument of the research. It is advisable to introduce a new section, designated as number 7, which should succinctly offer the conclusion in one or two paragraphs. This section should encapsulate the implications of the primary argument and its significance to the existing body of knowledge.

Author Response

First of all, I want to thank both reviewers for taking the time to engage with the text and provide valuable comments for the purpose of revision. In response to their feedback, I have made the following changes to the paper:

Both reviewers offered insights on the concluding discussion of the paper, albeit in somewhat different ways. Review 1 suggested the inclusion of a new paragraph with the primary findings and further highlighting the conclusion and significance to the existing body of knowledge. Reviewer 2 recommended more direct responses to the research questions. In light of these suggestions, I have restructured section 6 and introduced a concluding paragraph that summaries the analysis.

Review 1 provided commentary on the abstract and the appearance of feminist posthumanism. This has been revised and I have included the relational ontology and the key notions of the article to enhance and clarify the theoretical take.  

Reviewer 2 Report

Comments and Suggestions for Authors

This paper has elements of originality and was very interesting to read. However, some parts of the paper could be made clearer. 

In the introduction, the author states that the aims of the paper are:

- to theoretically and empirically explore how participatory methodologies involves affectivity.

- to illustrate how sexuality education offers specific affective conditions for our collective work.

Affectivity is therefore a key concept to the paper, and yet there is not much discussion or clarification on what 'affectivity' is, from a theoretical perspective. This could have been explored in more detail in section two, where the author discusses the theoretical underpinnings of the paper. 

The author's research questions include:

- how does affectivity induce the conditions of participatory methodologies, and

- how to address the (im)possibilities of the indeterminate character of affectivity in this setting?

These questions, particularly the second question, could have been answered more directly in section 6 of the paper. 

The methodology of the research project, namely why the research was being carried out with teachers and with what objectives, could also have been more clearly elucidated (apart from saying that the project was to reimagine what was doable and possible in sexuality education). I feel that with more information on why the research was being carried out, the author could show how recognition/addressing of affectivity could help to achieve those aims. 

Author Response

First of all, I want to thank both reviewers for taking the time to engage with the text and provide valuable comments for the purpose of revision. In response to their feedback, I have made the following changes to the paper:

Both reviewers offered insights on the concluding discussion of the paper, albeit in somewhat different ways. Review 1 suggested the inclusion of a new paragraph with the primary findings and further highlighting the conclusion and significance to the existing body of knowledge. Reviewer 2 recommended more direct responses to the research questions. In light of these suggestions, I have restructured section 6 and introduced a concluding paragraph that summaries the analysis.

Reviewer 2 requested a more comprehensive exploration of the concept of affectivity. This has been revised, see especially p. 6 (lines 144-154).

Furthermore, in response to Reviewer 2's request for additional information regarding the rationale behind the research, I have included pertinent details on pages 4 (lines 181-189) and 5 (lines 232-239).